# Medical Image Authentication Method Based on the Wavelet Packet and Energy Entropy

**DOI:** 10.3390/e24060798

**Published:** 2022-06-08

**Authors:** Tiankai Sun, Xingyuan Wang, Kejun Zhang, Daihong Jiang, Da Lin, Xunguang Jv, Bin Ding, Weidong Zhu

**Affiliations:** 1Faculty of Electronic Information and Electrical Engineering, Dalian University of Technology, Dalian 116024, China; strongtiankai@163.com; 2School of Information and Electrical Engineering, Xuzhou University of Technology, Xuzhou 221008, China; kej_zhang@163.com (K.Z.); daihong69@163.com (D.J.); lin_da76@163.com (D.L.); xg_jv66@163.com (X.J.); dingbin80@163.com (B.D.); wd_zhu69@163.com (W.Z.); 3School of Information Science and Technology, Dalian Maritime University, Dalian 116026, China

**Keywords:** wavelet packet decomposition, energy entropy, authentication, robustness

## Abstract

The transmission of digital medical information is affected by data compression, noise, scaling, labeling, and other factors. At the same time, medical data may be illegally copied and maliciously tampered with without authorization. Therefore, the copyright protection and integrity authentication of medical information are worthy of attention. In this paper, based on the wavelet packet and energy entropy, a new method of medical image authentication is designed. The proposed method uses the sliding window to measure the energy of the detail information. In the time–frequency data distribution, the local details of the data are mined. The complexity of energy is quantitatively described to highlight the valuable information. Based on the energy weight, the local energy entropy is constructed and normalized. The adjusted entropy value is used as the feature vector of the authentication information. A series of experiments show that the authentication method has good robustness against shearing attacks, median filtering, contrast enhancement, brightness enhancement, salt-and-pepper noise, Gaussian noise, multiplicative noise, image rotation, scaling attacks, sharpening, JPEG compression, and other attacks.

## 1. Introduction

With the widespread application of digital diagnosis technology, hospitals are producing a large amount of digital medical information every day. The digital medical information includes all kinds of medical images, electronic medical records (EPRs), electronic health records (EHRs), and diagnosis data. The rapid development of image analysis, signal processing, and 5G network technology have provided technical support for telemedicine, telemedicine consultation, and telemedicine teaching. Relying on efficient information processing methods, medical experts in different places can discuss diseases in real time. The application of new technology has further improved the diagnosis and treatment effect for patients. 

In the public network environment, the dissemination of digital medical information not only provides convenience for patients but also brings new security risks. When increasing amounts of medical data are transmitted to medical centers, the confusion of the medical data of different patients may cause medical negligence. Therefore, the medical data of different patients require extra care in the storage and distribution processes. At the same time, during the process of network transmission, any minor changes in medical information may create new medical disputes. Therefore, the security authentication of medical information is particularly important. In recent years, experts and scholars have studied a variety of medical image authentication schemes [1,2]. Some researchers have focused their efforts on detecting whether the medical images have been tampered. Others focused their efforts on identifying the tampered areas and repairing the tampered areas [3,4]. However, from the perspective of practical application, the authentication process of medical images should focus on whether the medical images have been tampered. The process should also focus on whether the authentication process impacts the medical diagnosis. In order to avoid distortion, some researchers divided the medical images into blocks; that is, the images were divided into regions of interest (ROIs) and regions of noninterest (RONIs). Different authentication methods are used for the ROI region and the RONI region. This kind of method is simple to implement, easy to operate, and easy to recover. The disadvantage of this scheme is that the method has poor fault tolerance and is easy for attackers to recover [1,5,6]. Some scholars have integrated different kinds of transformation domain operations such as DWT, DCT, DFT, quantization index modulation under dither modulation (QIM-DM), SVD, neural networks, and cryptography to authenticate medical images [7,8,9]. Several classical authentication schemes integrate various transform domain operations. For example, Anand integrated ECC encryption and digital watermarking technology to protect the security of medical information [3]. Singh used Hamming error correction code, a digital watermark, and cryptography to improve the security of authentication methods and reduce the bandwidth redundancy [8]. Thakur designed a double watermark model suitable for medical image security authentication by comprehensively using DWT, SVD, Hamming code, and chaotic encryption [9]. Jinhua designed a quantization-based image watermarking scheme by using the information entropy of the wavelet domain [10]. Hu et al. used a key to control the logistic chaotic map, and searched for the same binary sequence as the watermark information to realize authentication [11]. Based on DWT and visual cryptography, Hsieh and Huang proposed an authentication scheme, which is characterized by the mean and variance of the wavelet coefficients [12]. Aiming at the security protection of telemedicine applications, scholars such as Borra, Pirbhulal, and Farhan fully integrated FRT-SVD and cryptographic methods to realize the authentication of medical information [13,14,15,16]. Scholars such as Hsu and Hou have studied some security authentication methods by integrating cryptography and zero watermark. The implementation of such methods requires the integration of special transformation operations [17,18,19,20]. Compared with the spatial domain methods, the transformation domain methods have strong reversibility, high security, good visual quality, and resist attack. The disadvantages are that the calculation is complex and the implementation of the algorithm needs a specially designed scheme.

The effectiveness of all of the authentication schemes has been confirmed, and the different authentication methods each have their own advantages and disadvantages. Under the application background, the authentication model should ensure the uniqueness of data sources. The authentication model should also ensure the one-to-one correspondence between the data information and the relevant patients. A good authentication model should have good invisibility and robustness. Invisibility means that the authentication process cannot affect the quality of medical information. At the same time, the authentication process should not cause new medical disputes. Robustness means that the authentication model is still available under various geometric attacks and noise attacks. On the basis of previous work, we discussed the problem from the perspective of energy entropy. By using energy entropy, a suitable method for medical image authentication was designed. The innovations of this method are as follows:(1)In the time–frequency data distribution, the energy of the detailed information is described. Then, the complexity weight of the energy is measured. Based on this, the local energy entropy is constructed.(2)From the perspective of energy, the local details of the data are mined and then the local energy entropy is normalized. The processed entropy is used as the feature of the authentication information.(3)The proposed authentication method combines the advantages of multiresolution analysis and the stability of local energy entropy. No noise is added in the authentication process. The integrity goal of the authentication is achieved.

A series of attack experiments verified that the proposed method is robust in image compression, channel noise, as well as against intentional and unintentional attacks.

## 2. Basic Theory

### 2.1. Basic Theory of Wavelet Packet Transform

The wavelet packet transform is an extension of the wavelet transform, and its theory and algorithm are based on the wavelet transform [21,22,23]. The wavelet transform adopts a tower-type signal decomposition mode; that is, the signal is continuously decomposed on the low-frequency channel. The wavelet transform only extracts the low-frequency components and ignores the middle- and high-frequency features that may reflect the important information in the signal. The wavelet packet transform can gather in all frequency ranges, which not only retains the multiresolution characteristics but also makes full use of the rich detail information.

In orthogonal wavelet decomposition, through multiresolution analysis, only the subspace *V_j_* is decomposed into mutually orthogonal subspaces *V_j_*_+1_ and *W_j_*_+1_, namely:(1)$$V_{j}^{}=V_{j+1}^{}⊕W_{j+1}^{}$$

Different from the wavelet transform, the wavelet packet transform also further decomposes *W_j_* at any decomposition level. Assuming that the decomposition starts from *V*_0_, let $W_{0}^{0}=V_{0}^{}$, $\psi_{0}^{0}(t)=\phi (t)$, where ϕ(t) is the scaling function, then ${\left\{\psi_{0}^{0}(t-k)\right\}}_{k\in Z}$ is the orthonormal basis of $W_{0}^{0}$. First, $W_{0}^{0}$ is decomposed into $W_{1}^{0}$ and $W_{1}^{1}$, and there are $W_{1}^{0}⊥W_{1}^{1}$ and $W_{0}^{0}=W_{1}^{0}⊕W_{1}^{1}$. The subspace $W_{j}^{n}$ is decomposed into $W_{j+1}^{2n}$ and $W_{j+1}^{2n+1}$, and there are $W_{j+1}^{2n}⊥W_{j+1}^{2n+1}$ and $W_{j}^{n}=W_{j+1}^{2n}⊕W_{j+1}^{2n+1}$.Then, the orthonormal bases of subspaces $W_{j}^{n}$, $W_{j+1}^{2n}$, and $W_{j+1}^{2n+1}$ are ${\left\{\psi_{j}^{n}(t-2^{j}k)\right\}}_{k\in Z}$, ${\left\{\psi_{j+1}^{2n}(t-2^{j+1}k)\right\}}_{k\in Z}$, and ${\left\{\psi_{j+1}^{2n+1}(t-2^{j+1}k)\right\}}_{k\in Z}$, respectively, and they satisfy the following two-scale equations:(2)$$\{\begin{array}{c} \psi_{j+1}^{2n}(t)=\sum_{k\in z}h_{k}\psi_{j}^{n}(t-2^{j}k)\\ \psi_{j+1}^{2n+1}(t)=\sum_{k\in z}g_{k}\psi_{j}^{n}(t-2^{j}k) \end{array}$$
where $h_{k}$ and $g_{k}$ are a pair of conjugate mirror filters, and $g_{k}={\left(-1\right)}^{k}h_{1-k}$.

Let $f(t)\in V_{0}^{}$; according to the decomposition relationship of $V_{0}^{}$, for any specified layer *j*, $V_{0}^{}=W_{j}^{0}⊕W_{j}^{1}⊕\cdots ⊕W_{j}^{2^{j}-1}$.

Using the known filter $\left\{h_{k},g_{k}\right\}$ and the projection coefficient $d_{j,k}^{n}$ of *f* in subspace $W_{j}^{n}$ at scale *j*, the projection coefficients $d_{j+1,k}^{2n}$, $d_{j+1,k}^{2n+1}$ of *f* in subspaces $W_{j+1}^{2n}$ and $W_{j+1}^{2n+1}$ at scale *j* + 1 are calculated.

The decomposition and reconstruction algorithm of the wavelet packet are as follows:

Equation (3) gives the decomposition algorithm of the wavelet packet:(3)$$\{\begin{array}{c} d_{j+1,k}^{2n}=\sum_{l}h_{l-2k}d_{j,l}^{n}\\ d_{j+1,k}^{2n+1}=\sum_{l}g_{l-2k}d_{j,l}^{n} \end{array},k\in Z$$

Equation (4) gives the reconstruction algorithm of the wavelet packet:(4)$$d_{j,k}^{n}=\sum_{m}(h_{k-2m}d_{j+1,m}^{2n}+g_{k-2m}d_{j+1,m}^{2n+1}),k\in Z$$

### 2.2. Wavelet Packet Decomposition of Images

The wavelet decomposition of the image can be transformed into the wavelet decomposition of the one-dimensional signal (row and column). For the two-dimensional image f of M×N, the following steps can be used for wavelet packet decomposition:

(1) Wavelet transform is performed on image f, then four subimages $f⊗(H_{r},H_{c})$, $f⊗(H_{r},G_{c})$, $f⊗(G_{r},H_{c})$, and $f⊗(G_{r},G_{c})$ are obtained.

Among them, $f⊗(H_{r},H_{c})$ is a low-frequency image. $f⊗(H_{r},G_{c})$, $f⊗(G_{r},H_{c})$, and $f⊗(G_{r},G_{c})$ are the vertical, horizontal, and diagonal subimages, respectively, which are high-frequency images.

(2) Continue the operation in step (1) for the four subgraphs and then obtain the four subgraphs of each subimage. The cycle is carried out until *n*-level wavelet packet decomposition subgraphs are obtained.

Figure 1 shows the subgraphs of the image decomposed by the first- and second-level wavelet packets. 

### 2.3. Theory of Energy Entropy

Information entropy is a concept used to measure the amount of information in information theory [10]. It was proposed by Shannon in 1948 and is defined as follows:

Suppose that the information source X is a discrete random variable and the value of X is $X=\{x_{1},x_{2},\cdots ,x_{n}\}$. If the probability of the occurrence of each message is $P=\{p_{1},p_{2},\cdots ,p_{n}\}$ and $\sum_{i=1}^{n}p_{i}=1$, then the information entropy of X can be expressed as Formula (5):(5)$$H(X)=-\sum_{i=1}^{n}p_{i}\log p_{i}$$

Suppose that a two-dimensional image f is decomposed by wavelet packet. The energy corresponding to the *i*th sub-image $f_{i,}{}_{j}$ of the *j*th layer is denoted as $E_{i,}{}_{j}$, then
(6)$$E_{i,j}=\sum_{k=1}^{M^{\prime }}\sum_{l=1}^{N^{\prime }}{\left|f_{i,j}(k,l)\right|}^{2},\;i=0,1,\cdots ,4^{j}-1$$
where $M^{\prime }$ and $N^{\prime }$ are the size of the subimage $f_{i,}{}_{j}$, and $f_{i,}{}_{j}(k,l)$ represents the gray value of the subimage $f_{i,}{}_{j}(k,l)$ at (k,l). Then, the total energy of the *j*th layer is $E_{j}=\sum_{i=0}^{4^{j}-1}E_{i,j}$.

Let $P_{i,j}=\frac{E_{i,j}}{E_{j}}=\frac{E_{i,j}}{\sum_{i=0}^{4^{j}-1}E_{i,j}}$, then $\sum_{i=0}^{4^{j}-1}P_{i,j}=1$. According to information entropy theory, the energy entropy of the *i*th subimage $f_{i,}{}_{j}$ is defined as follows:(7)$$E_{WPEE}=-\sum_{i=0}^{4^{j}-1}P_{i,j}\log P_{i,j}$$

## 3. Authentication Scheme

### 3.1. Ownership Construction Phase

Multiresolution decomposition of medical images is carried out by the wavelet packet transform. In a time–frequency data distribution, a sliding window is used to measure the energy of the detailed information. Based on the energy weight, the local energy entropy is constructed. From the perspective of energy entropy, the local details of the data are mined. The local energy entropy is normalized and stored in a third-party certification center as authentication information. Figure 2 shows a schematic diagram of the copyright construction process. The detailed implementation process is shown below.


**Step 1: Extraction of robust features.**


When the robust areas of carrier information are fully used, the authentication method can better resist various attacks. The medical image *R* is multiscale-decomposed by the wavelet packet. After this operation, the low-frequency approximate wavelet coefficient *f*_11_ can be obtained. The low-frequency data *f*_11_ are decomposed into nonoverlapping blocks. The energy entropy of the segmented data can be calculated by Formulas (6) and (7) in Section 2.3.
[*f*_11_*,f*_12_*,f*_21_*,f*_22_] = DWT(R).


**Step 2: Construct feature vector.**


According to the energy entropy of each data block, the average energy entropy of the segmented data can be calculated. By using Equation (8), the relationship between the energy entropy of each segmented data and the mean energy entropy can be analyzed, and then the binary feature vector can be constructed.
(8)$$F=\left\{ \begin{array}{l} 0,\;if\;En_{i}>=En\\ 1,\;\mathrm{else} \end{array}\right.$$
where i=1,2,⋯,n×n, $En_{i}$ is the energy entropy of each block, and En is the mean energy entropy of all information blocks. The feature vector *F* is the original feature information extracted from the medical images by using wavelet packet energy entropy as the analysis tool.


**Step 3: Form authentication information and store authentication results.**


Perform the *XOR* operation between the copyright authentication information *IM* and the extracted feature vector *F*. *T*he results are stored in a third-party center for authentication.
(9)DW=XOR(IM,F)
where *IM* is the copyright logo data.

### 3.2. Ownership Verification Phase

Assuming that *R*’ is the image to be authenticated after a series of attacks, Figure 3 shows the schematic diagram of the authentication process. The detailed authentication process is shown below.


**Step1: Extraction of robust features.**


The suspected medical image *R′* is multiscale-decomposed by the wavelet packet. The low-frequency approximate wavelet coefficient *f*_11_*′* can be obtained. Then, the low-frequency data *f*_11_*′* are decomposed into nonoverlapping blocks. The energy entropy of the segmented data can be calculated by the Formulas (6) and (7) in Section 2.3.
[*f*_11_*′,f*_12_*′,f*_21_*′,f*_22_*′*] = *DWT*(*R′*).


**Step 2: Construct feature vector.**


According to the energy entropy of each data block, the average energy entropy of the segmented data is calculated. By using Equation (10), the relationship between the energy entropy of each segmented data and the mean energy entropy is analyzed. The binary feature vector *F′* can be constructed.
(10)$$F^{\prime }=\left\{ \begin{array}{l} 0,\;if\;E^{\prime }n_{i}>=E^{\prime }n\\ 1,\;\mathrm{else} \end{array}\right.,$$
where i=1,2,⋯,n×n,$E^{\prime }n_{i}$ is the energy entropy of each block, and $E^{\prime }n$ is the mean energy entropy of all the information blocks. The feature vector *F′* is the feature information extracted from the suspected images.


**Step3: Complete verification.**


Perform the XOR operation between the feature information *F′* and *DW*, which is stored in a third-party authentication center. The results are stored in matrix *IM′* and *IM′* is the recovered authentication information. The authentication is completed by judging the difference between *IM* and *IM′*.
(11)$$IM^{\prime }=XOR(DW,F^{\prime })$$
where *IM* is the original logo data, and *IM′* is the recovered logo data.

## 4. Analysis of Experimental Results

In order to verify the feasibility of this method, six different types of medical images were selected as the test images from the medical image platform at https://peir.path.uab.edu/library/We, accessed the platform, (accessed on 10 August 2021). The six images were from different parts of the human body, which better verified the robustness of the proposed method. Figure 4 gives the test images and a 32 × 32 logo image. The proposed method was tested in the six images. Due to the space limitations, we provide the test results on breast images and presents the averaged results for all six test images.

The peak signal-to-noise ratio (*PSNR*) of the original image and the attacked image is used to quantitatively describe the impact of various attacks on the original carrier information. The greater the *PSNR* value, the higher the similarity of the two images. When the *PSNR* value is less than 30 dB, the human eye can perceive the difference between the original image and the attacked image. The evaluation index is more consistent with the visual perception characteristics of the human eyes. The robustness of the authentication method is evaluated by the normalized similarity value (*NC*). The *NC* value is between zero and one. The larger the *NC* value, the higher the similarity between them. The robustness of the authentication method can be verified when the *PSNR* is very low and the *NC* value is very high.
(12)$$\begin{array}{l} PSNR=10\log_{10}\frac{255^{2}}{MSE}(dB),\\ MSE=\frac{1}{M\times N}\sum_{i=1}^{M}\sum_{j=1}^{N}{\left(H_{ij}-{H^{\prime }}_{i}{}_{j}\right)}^{2}, \end{array}$$
where *M* and *N* are the width and height of the tested image; $H_{ij},{H^{\prime }}_{ij}^{}$ are the pixel value of the carrier image before and after the attack, respectively.
(13)$$NC=1-\frac{\sum_{b=1}^{B}w_{b}⊕{w^{\prime }}_{b}}{B}$$
where $w_{b},{w^{\prime }}_{b}^{}$ are the original authentication information and the extracted copyright information, respectively; *B* is the size of the copyright information.

### 4.1. Correlation Test between Different Features

The features used for authentication should have strong autocorrelation with the carrier. The authentication features extracted from different images should have strong independence. The authentication features extracted from different images should be independent of each other. In order to verify the autocorrelation between the features, Table 1 is used to show the correlation values of the different features. The correlation between different features is evaluated by the normalized similarity value (*NC*). The *NC* value is between zero and one. The larger the *NC* value, the higher the similarity between them. It can be seen from Table 1 that most of the data are close to 0.7 and some are close to 0.3. The results indicate that the features extracted from different images were different.

### 4.2. Analysis of Experimental Results

Due to the interference of network noise and human factors, medical information may change in the process of network transmission. In order to intuitively verify the effectiveness of the proposed method, we use Figure 5, Figure 6, Figure 7, Figure 8, Figure 9, Figure 10, Figure 11, Figure 12, Figure 13, Figure 14 and Figure 15, which show the robustness results of a breast after a series of simulated attacks. The simulated attack types were image rotation (rotation 10 degrees), scaling attack (zoom to 200%), sharpening attack, JPEG compression attack (compression factor 10%), salt and pepper noise attack (parameter 0.001), Gaussian noise attack (Gaussian parameter 0.005), multiplicative noise attack (noise parameter 0.01), contrast enhancement attack, brightness enhancement attack, clipping attack (cutting out 1/5 of the original image), and median filtering attack (filtering parameter 5 × 5).

Figure 5, Figure 6, Figure 7, Figure 8, Figure 9, Figure 10, Figure 11, Figure 12, Figure 13, Figure 14 and Figure 15 show the robustness of the breast image after a series of simulated attacks. Table 2 presents the averaged results of the six test images.

Medical information is different from traditional data information. Even slight changes in medical information may cause medical disputes. From Figure 5, Figure 6, Figure 7, Figure 8, Figure 9, Figure 10, Figure 11, Figure 12, Figure 13, Figure 14 and Figure 15, it can be seen that the proposed method has good robustness against conventional attacks, especially against JPEG compression (compression factor 10%), rotation attack (rotation 10 degrees), scaling attack (zoom in 200%), multiplicative noise (noise parameter 0.01), brightness enhancement, etc. The *NC* value of the authentication image was one. The *NC* values of the other attack tests were also close to one. Especially for the clipping attack, the visual change was obvious after the attack (*PSNR* = 15.7723 after the attack). The *NC* value of the authentication information extracted from the attacked image was still close to one (*NC* = 0.9688). After Gaussian noise, contrast enhancement, and brightness enhancement attacks, the visual change in the original medical image was also obvious (*PSNR* value was less than 30). The *NC* value of the authentication information extracted from the attacked image was also close to one. It can be seen that the proposed method has good robustness against both conventional geometric and nongeometric attacks.

### 4.3. Algorithm Comparison Test

Figure 5, Figure 6, Figure 7, Figure 8, Figure 9, Figure 10, Figure 11, Figure 12, Figure 13, Figure 14 and Figure 15 and Table 2 verify the effectiveness of the method from both visual effects and numerical calculations. To further verify the robustness of the proposed method, Table 3, Table 4 and Table 5 and Figure 16 and Figure 17 show the comparison results between this method and other authentication methods.

Table 4 and Figure 16 describe the comparison results with reference methods [11,18,19,20] under various attacks such as brightness, median filtering, scaling change, noise, rotation, JPEG compression, clipping, etc. The results show that the proposed method has good robustness under various attacks. From Figure 16, compared with the reference methods [11,18,19,20], the proposed method shows good robustness in resisting Gaussian noise, median filtering, JPEG compression, rotation attack, etc. The robustness of the proposed method against cropping attack is slightly weaker than that of the reference methods [11,18,19,20]. The proposed method can meet security authentication requirements.

In comparison with reference methods [3,8,9], the proposed method shows good robustness against salt-and-pepper noise, Gaussian noise, multiplicative noise, rotation attack, JPEG compression, scaling, median filtering, etc. From Table 5 and Figure 17, the *NC* value of seven attack tests is equal to 1 and the other NC values are also close to 1.

Through comparative analysis with previously reported methods [3,8,9,11,18,19,20], by mining and analyzing the local features of wavelet packet energy entropy, the formed authentication information shows good robustness in resisting Gaussian noise, median filtering, JPEG compression, rotation attack, and so on. Limited by the aggregation of energy, the robustness of the proposed method against cropping attack is slightly weaker.

## 5. Conclusions

Without adding any noise or making any change to the original medical image, based on the wavelet packet energy entropy, the complexity of the information energy was quantitatively described. From the perspective of local energy, the features of the data were fully mined, and the local energy entropy was constructed. By using the energy measurement the feature vector was formed. A series of attack tests showed that the authentication method has good rotation invariance and scale invariance. The results showed strong robustness against common geometric deformation and various kinds of noise attacks. At the same time, this method has good universality and is especially useful for military images and medical images.

## Figures and Tables

**Figure 1 entropy-24-00798-f001:**
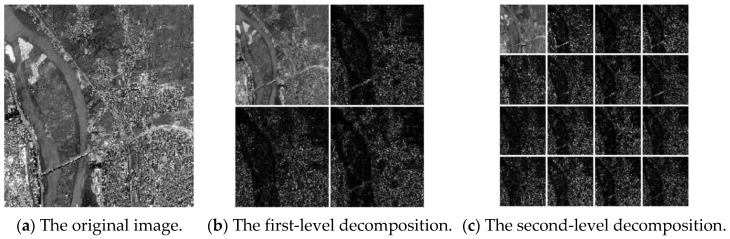
An example of two-level wavelet packet decomposition. (**a**) represents the original image, (**b**) represents the 4 subimages of the first-level wavelet packet decomposition, and (**c**) represents the 16 subimages of the second-level wavelet packet decomposition.

**Figure 2 entropy-24-00798-f002:**
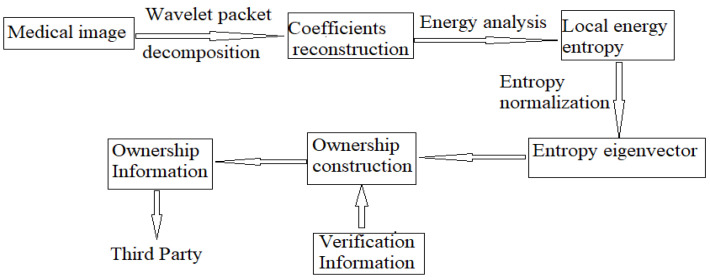
Flow chart of the copyright construction.

**Figure 3 entropy-24-00798-f003:**
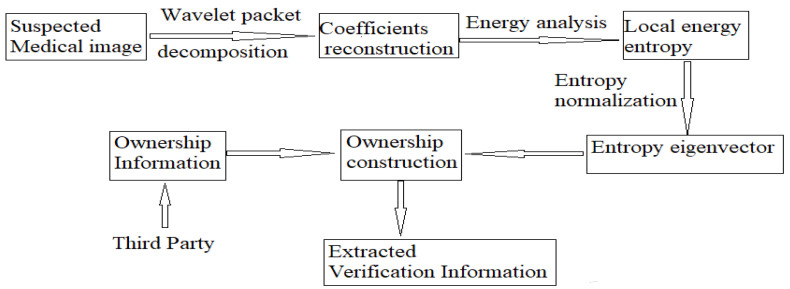
Flow chart of the ownership verification.

**Figure 4 entropy-24-00798-f004:**
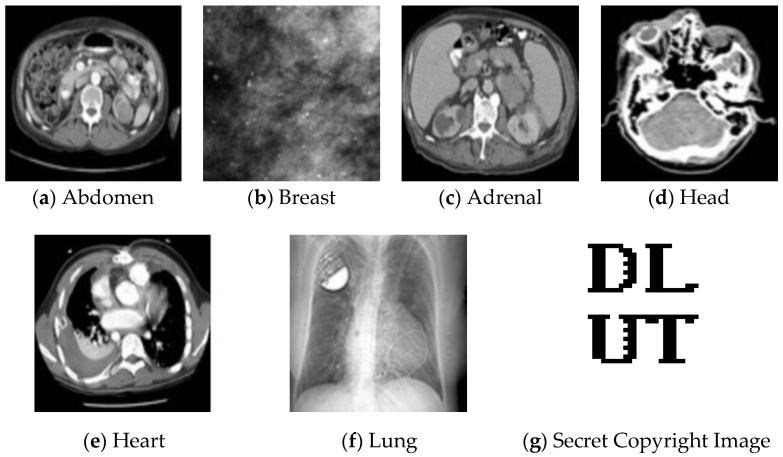
The test images and copyright logo.

**Figure 5 entropy-24-00798-f005:**
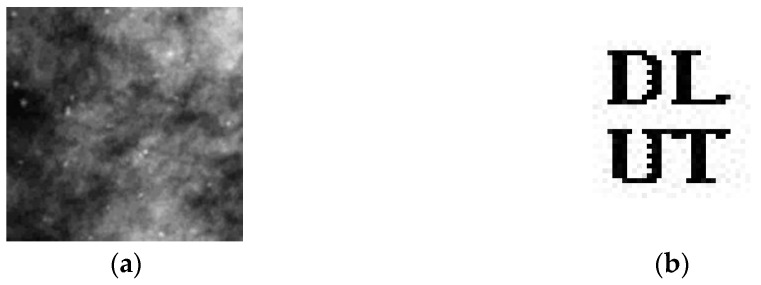
(**a**) JPEG compression attack PSNR = 34.4752; (**b**) recovered logo image from (**a**) (NC = 1).

**Figure 6 entropy-24-00798-f006:**
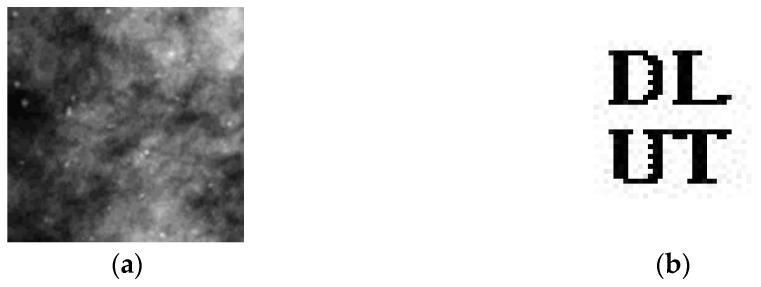
(**a**) Salt and pepper noise attack PSNR = 35.1107; (**b**) recovered logo image from (**a**) (NC = 0.9980).

**Figure 7 entropy-24-00798-f007:**
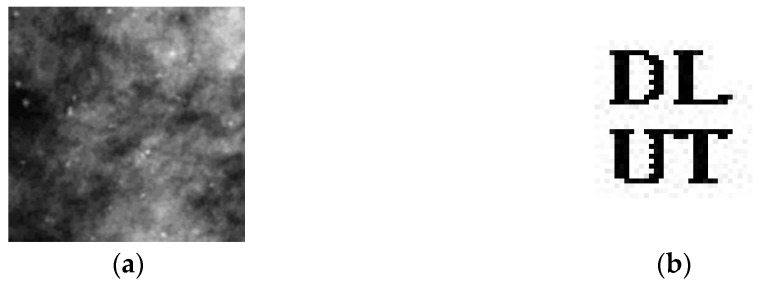
(**a**) Gaussian noise attack PSNR = 22.4612; (**b**) recovered logo image from (**a**) (NC = 0.9824).

**Figure 8 entropy-24-00798-f008:**
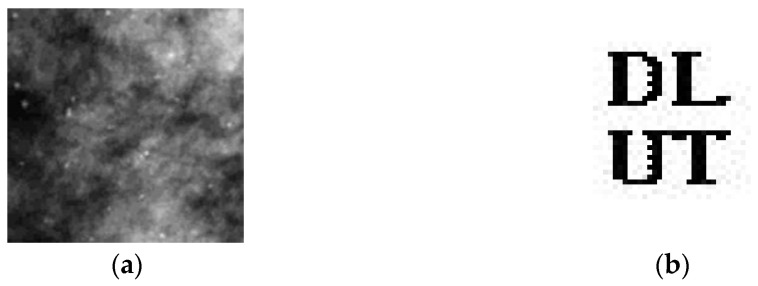
(**a**) Rotation attack PSNR = 34.7317; (**b**) recovered logo image from (**a**) (NC = 1).

**Figure 9 entropy-24-00798-f009:**
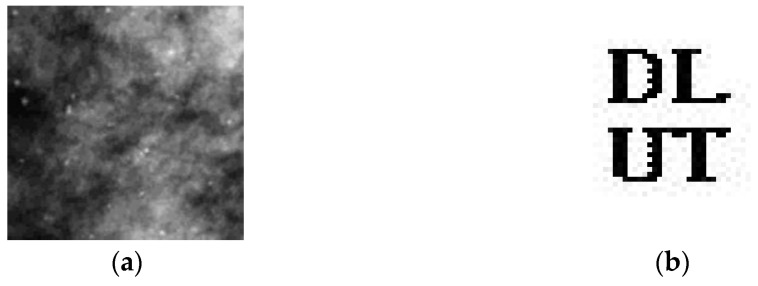
(**a**) Scaling attack PSNR = 56.0139; (**b**) recovered logo image from (**a**) (NC = 1).

**Figure 10 entropy-24-00798-f010:**
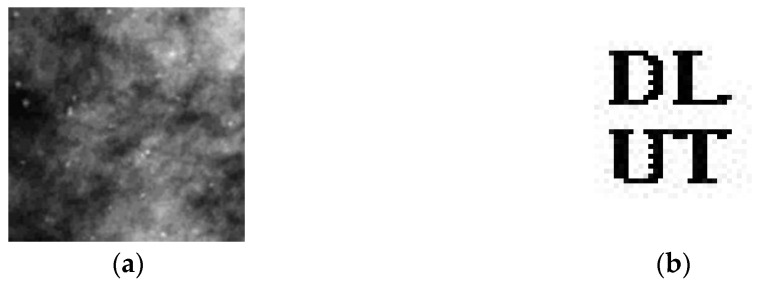
(**a**) Sharpening attack PSNR = 30.6776; (**b**) recovered logo image from (**a**) (NC = 0.9990).

**Figure 11 entropy-24-00798-f011:**
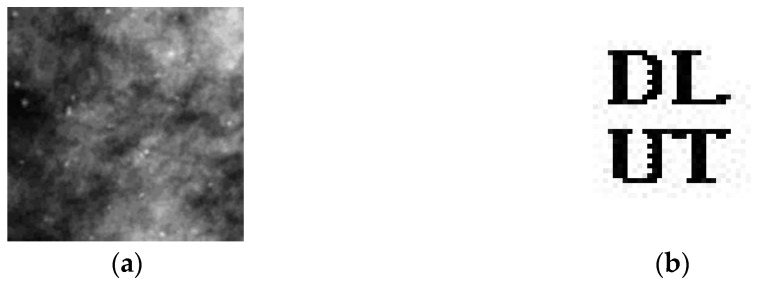
(**a**) Multiplicative noise attack PSNR = 28.4653; (**b**) recovered logo image from (**a**) (NC = 1).

**Figure 12 entropy-24-00798-f012:**
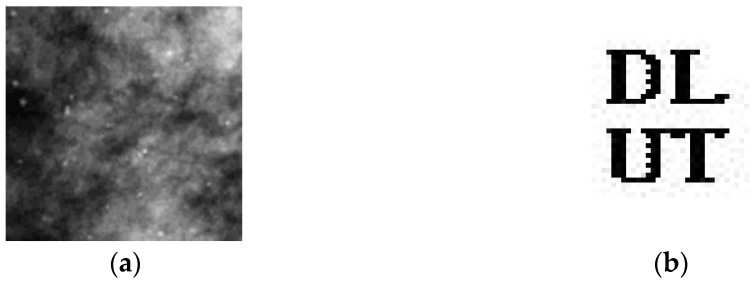
(**a**) Clipping attack PSNR = 15.7723; (**b**) recovered logo image from (**a**) (NC = 0.9688).

**Figure 13 entropy-24-00798-f013:**
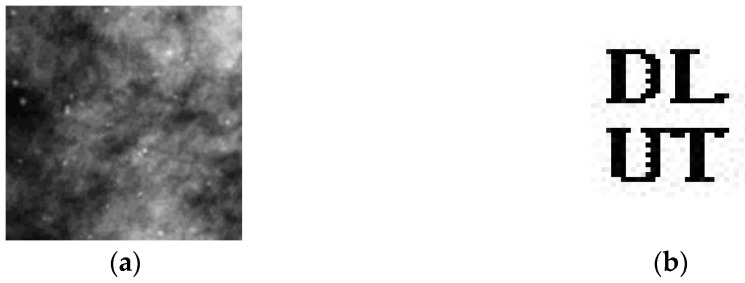
(**a**) Median filtering attack PSNR = 43.9290; (**b**) recovered logo image from (**a**) (NC= 0.9990).

**Figure 14 entropy-24-00798-f014:**
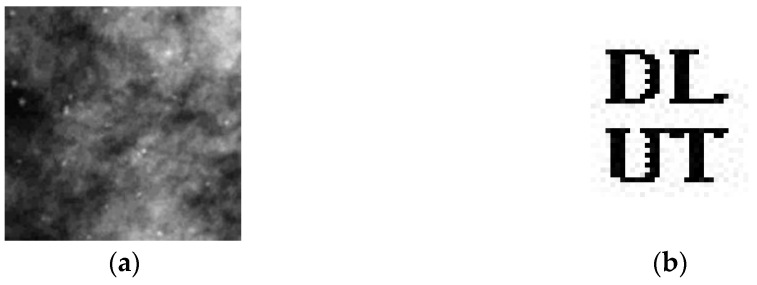
(**a**) Contrast enhancement attack PSNR = 21.5881; (**b**) recovered logo image from (**a**) (NC = 0.9766).

**Figure 15 entropy-24-00798-f015:**
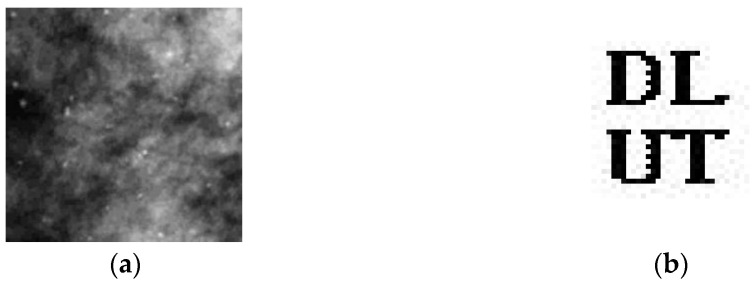
(**a**) Brightness enhancement attack PSNR = 28.2620; (**b**) recovered logo image from (**a**) (NC = 1).

**Figure 16 entropy-24-00798-f016:**
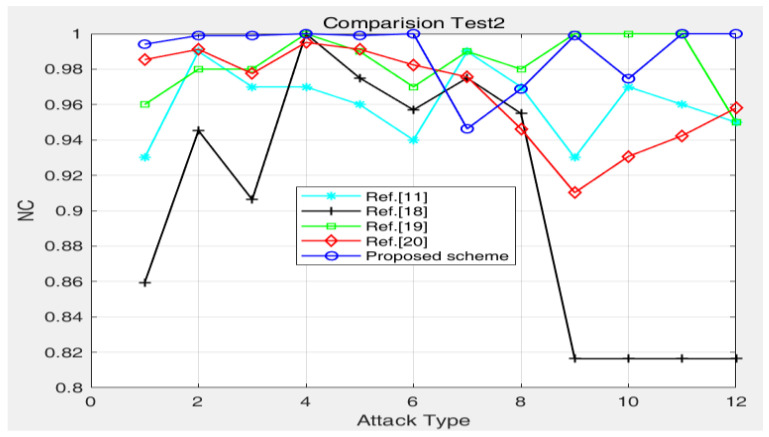
Comparisons of the robustness between different schemes: (1) attacks type 1–12 are Gaussian noise, median filtering 5 × 5, median filtering 7 × 7, JPEG (70%), JPEG (50%), JPEG (20%), cropping (10%), cropping (20%), rotation attack (1 degree), rotation attack (2.5 degree), rotation attack (5 degrees), and rotation attack (10 degrees), respectively.

**Figure 17 entropy-24-00798-f017:**
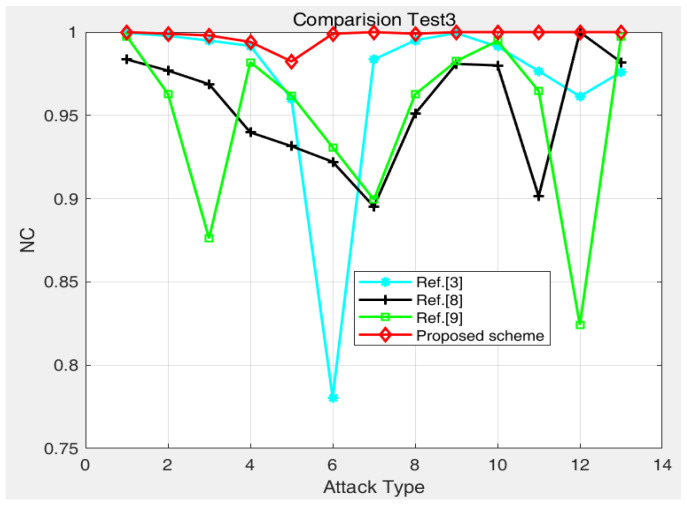
Comparisons of the robustness between different schemes: (2) attacks type 1–13 are salt and pepper noise (0.0001), salt and pepper noise (0.0005), salt and pepper noise (0.001), Gaussian noise (0.001), Gaussian noise (0.005), rotation (1 degree), JPEG (10%), JPEG (50%), JPEG (90%), speckle noise (0.001), speckle noise (0.005), image scaling (2×), and median filtering 1 × 1), respectively.

**Table 1 entropy-24-00798-t001:** Similarity test of different features.

	Abdomen (b) Breast (c) Chest (d) Head	Breast	Adrenal	Head	Heart	Lung
Abdomen	1	0.2793	0.7334	0.7490	0.6777	0.7188
Breast	0.2793	1	0.1533	0.2412	0.2891	0.0156
Adrenal	0.7334	0.1533	1	0.7578	0.7471	0.8467
Head	0.7490	0.2412	0.7578	1	0.6748	0.7432
Heart	0.6777	0.2891	0.7471	0.6748	1	0.7246
Lung	0.7188	0.0156	0.8467	0.7432	0.7246	1

**Table 2 entropy-24-00798-t002:** Averaged results of six test images.

Attack	Breast	Averaged Results of Six Test Images
JPEG (10)	1	0.9952
Salt-and-pepper noise (0.001)	0.9980	0.9749
Gaussian noise (0.005)	0.9824	0.9819
Rotation (10 degrees)	1.0000	0.9968
Scale scaling (200%)	1	0.9924
Sharpening	0.9990	0.9938
Multiplicative noise (0.01)	1	0.9918
Clipping (20%)	0.9688	0.9637
Median filtering 5 × 5	0.9990	0.9970
Contrast enhancement	0.9766	0.9852
Brightness enhancement	1	0.9853

**Table 3 entropy-24-00798-t003:** Comparison tests (1).

Attacks	Hsieh and Huang’s Scheme [12]	Hsu and Hou’s Scheme [17]	Tiankai’s Scheme [20]	Proposed Scheme
Sharpening	0.752	0.819	0.9561	0.9990
Median filtering	0.843	0.938	0.9775	0.9990
Resizing	0.733	0.887	0.9521	1
Noise addition	0.723	0.761	0.9854	0.9941
JPEG	0.845	0.956	0.9912	0.9990

**Table 4 entropy-24-00798-t004:** Comparison tests (2).

Attack	Ref. [11]	Ref. [18]	Ref. [19]	Ref. [20]	Proposed Scheme
Gaussian noise	0.9300	0.8594	0.9600	0.9854	0.9941
Median filtering 5 × 5	0.9900	0.9453	0.9800	0.9912	0.9990
Median filtering 7 × 7	0.9700	0.9063	0.9800	0.9775	0.9990
JPEG (70)	0.9700	1.0000	1.0000	0.9951	1
JPEG (50)	0.9600	-	0.9900	0.9912	0.9990
JPEG (20)	0.9400	0.9570	0.9700	0.9824	1
Cropping (10%)	0.9900	-	-	0.9756	0.9463
Cropping (20%)	0.9700	-	-	0.9463	0.9688
Rotation attack (1 degree)	0.9300	0.8164	-	0.9102	0.9990
Rotation attack (2.5 degrees)	0.9700	-	-	0.9307	0.9746
Rotation attack (5 degrees)	0.9600	-	1.0000	0.9424	1
Rotation attack (10 degrees)	0.9500	-	0.9500	0.9580	1
Visibility	No	No	No	Yes	Yes

**Table 5 entropy-24-00798-t005:** Comparison test (3).

Attack	Noise Density	Ref. [3]	Ref. [8]	Ref. [9]	Proposed Scheme
Salt-and-pepper noise	0.0001	0.9995	0.9836	0.9975	1
Salt-and-pepper noise	0.0005	0.9977	0.9769	0.9630	0.9990
Salt-and-pepper noise	0.001	0.9949	0.9687	0.8761	0.9980
Gaussian noise	0.001	0.9917	0.9398	-	0.9941
Gaussian noise	0.005	0.9599	0.9315	-	0.9824
Rotation	1 degree	0.7806	0.9221	0.9308	0.9990
JPEG compression	QF = 10	0.9835	0.8952	0.8994	1
JPEG compression	QF = 50	0.9951	0.9510	0.9626	0.9990
JPEG compression	QF = 90	0.9994	0.9809	-	1
Speckle noise	0.001	0.9913	0.9800	0.9947	1
Speckle noise	0.005	0.9766	0.9014	-	1
Image scaling	2	0.9614	-	0.8242	1
Median filter	[11]	0.9760	0.9819	0.9973	1

## Data Availability

The medical images selected as the test images were taken from the medical image platform at https://peir.path.uab.edu/library/ (accessed on 10 August 2021).

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
