# Peer review of "Medical Image Authentication Method Based on the Wavelet Packet and Energy Entropy"

_entropy, 2022, doi:10.3390/e24060798_

Round 1

Reviewer 1 Report

I have checked the resubmitted manuscript and the cover letter. I am satisfied with the new version and I have no more comments.

Author Response

Point 1: I have checked the resubmitted manuscript and the cover letter. I am satisfied with the new version and I have no more comments.

 Response 1: Thank you for your previous review suggestions. Once again, I would like to express my heartfelt thanks for your hard work.

Reviewer 2 Report

First, the writing style/clarity needs more effort before this work is published. Indeed, the paper is challenging to read due to the poor grammar used throughout and the unclear structure of the argument being put across. In particular, the quality of the presentation significantly weakens this paper. Therefore, the language used in this paper needs to be run through a professional editor in order to make it easier to follow and comprehend. This paper would be substantially improved by thoroughly rewriting the prose with the help of a good English-language writer. In general, this paper needs such a treatment before being considered any further.

Furthermore, presentation aside, by reading the paper, it still was not entirely clear what to expect with the direction of the article. Indeed, the contribution proposed in this paper has been only marginally compared and contextualized with respect to the state-of-the-art. As a result, it is challenging to understand the novelty introduced by the paper. It is not clear which are the concrete advantages of the proposed method. Furthermore, it is unclear which problems (or gaps) will solve the authors' proposal, especially if compared to existing solutions. Again, the authors should provide more general motivations for the proposed method, besides providing additional usage scenarios. The aspects mentioned above should be carefully addressed before the paper can be considered any further.

The set of references should be expanded by taking into account the following related works:
- https://doi.org/10.1177%2F15501477211014132
- https://doi.org/10.1109/AINA.2015.224

As a minor remark, thorough proofreading is strongly recommended.

Author Response

Response to Reviewer Comments

Point 1: First, the writing style/clarity needs more effort before this work is published. Indeed, the paper is challenging to read due to the poor grammar used throughout and the unclear structure of the argument being put across. In particular, the quality of the presentation significantly weakens this paper. Therefore, the language used in this paper needs to be run through a professional editor in order to make it easier to follow and comprehend. This paper would be substantially improved by thoroughly rewriting the prose with the help of a good English-language writer. In general, this paper needs such a treatment before being considered any further.

Response 1:  

With the help of a professional editor, the writing has been improved and some long sentences have been revised. Some grammar and structure problems have been revised.

The added and modified parts are marked in red font.

 Point 2: Furthermore, presentation aside, by reading the paper, it still was not entirely clear what to expect with the direction of the article. Indeed, the contribution proposed in this paper has been only marginally compared and contextualized with respect to the state-of-the-art. As a result, it is challenging to understand the novelty introduced by the paper. It is not clear which are the concrete advantages of the proposed method. Furthermore, it is unclear which problems (or gaps) will solve the authors' proposal, especially if compared to existing solutions. Again, the authors should provide more general motivations for the proposed method, besides providing additional usage scenarios. The aspects mentioned above should be carefully addressed before the paper can be considered any further.

Response 2:  

By studying the reference you provided -https://doi.org/10.1109/AINA.2015.224.

We find that the core idea of our paper is to use zero watermarking technology to protect medical images.

The innovations of this method are as follows: Based on the wavelet packet and energy entropy, a new method of zero watermarkin is designed. The proposed method uses the sliding window to measure the energy of the detail information. In the time-frequency data distribution, the local details of the data are mined. The complexity of energy is described quantitatively so as to highlight the valuable information. Based on the energy weight, the local energy entropy is constructed and normalized. The adjusted entropy value is used as the feature vector of the authentication information.

In the experimental results part, two comparative line diagrams (FIG. 16 and FIG. 17) are added which better give intuitive explanations on the result. The added and modified parts are marked in red font.

Point 3: The set of references should be expanded by taking into account the following related works:
- https://doi.org/10.1177%2F15501477211014132
- https://doi.org/10.1109/AINA.2015.224

Response 3:  

We have carefully studied the core ideas of these two references. These two articles have been added to the references. All references have been revised according to your advise.

Point 4: As a minor remark, thorough proofreading is strongly recommended.

Response4: The writing has been improved and some long sentences have been revised. The added and modified parts are marked in red font.

 Thank you for your review suggestions. Once again, I would like to express my heartfelt thanks for your hard work.

Reviewer 3 Report

This paper describes a novel way to authenticate medical images by using a robust feature vector from the local energy entropy computed from the wavelet packets. The paper is well written and gives convincing results.

I have two suggestions for improving the paper:

In the results section 4, the proposed scheme is compared to previous schemes only with respect to a single "breast" image. I would advise the authors to rather present results which are averaged over all six test images.

In theory, the wavelet packet transform may be chosen in an arbitrary way (see e.g. Pommer, Andreas, and Andreas Uhl. "Selective encryption of wavelet-packet encoded image data: efficiency and security." Multimedia Systems 9.3 (2003): 279-287.) The authors might consider to use the wavelet packet decomposition as a secret parameter to further enhance security of the scheme.

Author Response

Response to Reviewer Comments

This paper describes a novel way to authenticate medical images by using a robust feature vector from the local energy entropy computed from the wavelet packets. The paper is well written and gives convincing results. I have two suggestions for improving the paper:

 Point 1: In the results section 4, the proposed scheme is compared to previous schemes only with respect to a single "breast" image. I would advise the authors to rather present results which are averaged over all six test images.

 Response 1:

One table has been added to present the averaged results over all six test images. Table 2 presents the averaged results over all six test images. The added and modified parts are marked in red font.

 Point 2: In theory, the wavelet packet transform may be chosen in an arbitrary way (see e.g. Pommer, Andreas, and Andreas Uhl. "Selective encryption of wavelet-packet encoded image data: efficiency and security." Multimedia Systems 9.3 (2003): 279-287.) The authors might consider to use the wavelet packet decomposition as a secret parameter to further enhance security of the scheme.

Response 2: By studying the reference "Selective encryption of wavelet-packet encoded image data: efficiency and security." Multimedia Systems 9.3 (2003): 279-287.) . We know that using the wavelet packet decomposition as a secret parameter to further enhance security of the scheme is a very useful method. In the following papers, we will further use your good ideas..

Thank you for your review suggestions. Once again, I would like to express my heartfelt thanks for your hard work.

Round 2

Reviewer 2 Report

The authors addressed all the issues I pointed out by carefully revising the paper. The paper has been substantially improved with respect to the relative previous version. Due to the reasons given above, the paper is now ready to be accepted for publication.

This manuscript is a resubmission of an earlier submission. The following is a list of the peer review reports and author responses from that submission.

Round 1

Reviewer 1 Report

The submitted paper suffers many substantial weakness so this reviewer recommends rejecting it. First, the title and the introduction promises an authentication method for certain (medical) images. Authentication is a method which ensures that the document (image) originates from the claimed source: the document is authentic. In this case a user trusts the document as fas as he/she trusts the issuer. Seemingly this is not what the paper is about. Integrity is another property as discussed lengthy in the Introduction. It makes sure that the document did not change, it is the same as has been originally created, it has not been tampered with. The industry standard for achieving both authenticity and integrity is digital signature: the issuer appends a digital signature to the hash of the document. Applying it solves all problems discussed in the Introduction.

In Section 4 the paper discusses that the suggested scheme is resistant to certain image manipulation. This possibility is in complete contrast with what has been said in the Introduction, where special emphasis was given for protecting the uniqueness of the image, up to small details.

What is discussed in Section 3 is a kind of feature extraction which is claimed to be robust against some image manipulation. The "ownership construction" (actually, adding some kind of watermark to the image, and storing the result at a trusted third party) is only one and a very bad application of feature extraction.  Rather than showing the mistakenly recovered bits on Figures 5--9 it is visually more spectacular to show some "copyright image", but this adds nothing to it. Feature extraction, however, is not what the first two sections and the title of the paper are about.

Even the description of that "feature extraction" has significant flaws.  By the examples, from an original image the method produces 1024 bits, arranged in a 32 by 32 zero-one matrix. If the reviewer deciphered the algorithm correctly, the bits are constructed as follows. Wavelet packet transform is applied to the image at depth 5 (to make a 32 by 32 subgrid of the original image). On each part an entropy-like amount is computed from the wavelet coefficients. Finally, the extracted bit shows if this amount is below or above the average. The description of wavelet packet transform is messy. On line 100 neither  h nor l is defined (what they are?). In formula (1) u0 appears on the left and the right hand side of the definition. Subspaces are never used later, why are they there? The decomposition algorithm is not used at all, why does it appear here? In the description of the actual algorithm (Section 2.2) important details are missing: how the image is padded (as the wavelet algorithm requires the image size to be a power of two); what basic wavelet functions are used (there are tens of standard possibilities, which one has been used, and why?) After getting the wavelet coefficients, entropy is computed. It is not justified why the authors decided to use this entropy -- which is typically used to truncate wavelet packet creation -- and not something completely different amount.

Author Response

Response to Reviewer 1 Comments

Point 1: First, the title and the introduction promises an authentication method for certain (medical) images. Authentication is a method which ensures that the document (image) originates from the claimed source: the document is authentic. In this case a user trusts the document as fas as he/she trusts the issuer. Seemingly this is not what the paper is about. Integrity is another property as discussed lengthy in the Introduction. It makes sure that the document did not change, it is the same as has been originally created, it has not been tampered with. The industry standard for achieving both authenticity and integrity is digital signature: the issuer appends a digital signature to the hash of the document. Applying it solves all problems discussed in the Introduction.

 Response 1: The transmission of digital medical information in an open network platform will be affected by data compression, noise, scaling, labeling and so on. At the same time medical data may be illegally copied and maliciously tampered with without authorization. Therefore, in the process of telemedicine, the copyright protection, integrity authentication of medical information are worthy of attention. At the same time during the process of network transmission any minor changes of medical information may create new medical disputes. Therefore, the security authentication of medical information in the process of network transmission is particularly important.

 Point 2: In Section 4 the paper discusses that the suggested scheme is resistant to certain image manipulation. This possibility is in complete contrast with what has been said in the Introduction, where special emphasis was given for protecting the uniqueness of the image, up to small details.

Response 2: Under the application background the authentication model should not only ensure the uniqueness of data sources but also be able to ensure the one-to-one correspondence between the data information and the relevant patients. A good authentication model must have good invisibility and robustness. Invisibility means that the authentication process cannot affect the quality of medical information, and the authentication process cannot cause new medical disputes. The robustness means that the authentication model is still available under various geometric attacks and noise. A series of attack experiments verify that the proposed method is robust against intentional or unintentional attacks.

Point 3: What is discussed in Section 3 is a kind of feature extraction which is claimed to be robust against some image manipulation. The "ownership construction" (actually, adding some kind of watermark to the image, and storing the result at a trusted third party) is only one and a very bad application of feature extraction.  Rather than showing the mistakenly recovered bits on Figures 5--9 it is visually more spectacular to show some "copyright image", but this adds nothing to it. Feature extraction, however, is not what the first two sections and the title of the paper are about. Even the description of that "feature extraction" has significant flaws.  By the examples, from an original image the method produces 1024 bits, arranged in a 32 by 32 zero-one matrix. If the reviewer deciphered the algorithm correctly, the bits are constructed as follows. Wavelet packet transform is applied to the image at depth 5 (to make a 32 by 32 subgrid of the original image). On each part an entropy-like amount is computed from the wavelet coefficients. Finally, the extracted bit shows if this amount is below or above the average.

Response 3: Any slight change of medical information may lead to new medical disputes. The proposed method is the idea of zero watermarking authentication. No noise is added in the authentication process and the integrity goal of the medical information authentication process is achieved. The transmission of digital medical information in an open network platform will be affected by data compression, noise, scaling, labeling and so on. At the same time medical data may be illegally copied and maliciously tampered with without authorization. Figures 5-15 are to verify that the authentication features have good robustness even after a series of intentional or unintentional attacks on the original medical image.

 Point 4: The description of wavelet packet transform is messy. On line 100 neither  h nor l is defined (what they are?). In formula (1) u0 appears on the left and the right hand side of the definition. Subspaces are never used later, why are they there? The decomposition algorithm is not used at all, why does it appear here? In the description of the actual algorithm (Section 2.2) important details are missing: how the image is padded (as the wavelet algorithm requires the image size to be a power of two); what basic wavelet functions are used (there are tens of standard possibilities, which one has been used, and why?) After getting the wavelet coefficients, entropy is computed. It is not justified why the authors decided to use this entropy -- which is typically used to truncate wavelet packet creation -- and not something completely different amount.

Response 4: The subsection of the basic theory of wavelet packet transform has been rewritten. The ‘bior3.7’ has been used as the tested wavelet in the experienment. This paper focuses on the stability of wavelet packet energy entropy feature. Which wavelet has better stability is not the focus of this paper. The added and modified parts are marked in red font.

Thanks again for your hard work.

Reviewer 2 Report

This paper proposes a medical image authentication method based on the wavelet p0acket and energy entropy. The proposed method is interesting, and I feel that this paper should undergo significant revision before being accepted for publication. Here are some comments that should be taken seriously.

  1. In the abstract, previous works on the topic of image authentication should be briefly reviewed to summarize the challenge in the field. Why the proposed method can help to solve the challenge should be explained.
  2. More previous works on image authentication should be presented in the introduction part, and should be analyzed and compared in the experimental section.
  3. Equations should appear as part of a sentence and should be punctuated accordingly. However, most of the equations presented in this manuscript are not treated as part of a sentence. The unprofessional mathematical writing significantly degrades the quality of this manuscript and leads to an unpleasant reading experience.
  4. The format of the reference is not consistent.
  5. My major concern is the novelty of this paper. It appears to me the contributions in this paper are existing well-established methods exploited in authentication. The novelty of the methodology is limited. The authors should justify the originality of the proposed method.
  6. The experimental results section should be well presented to show how the parameters affect the performance of the proposed method.

Author Response

Response to Reviewer 2 Comments

Point 1: In the abstract, previous works on the topic of image authentication should be briefly reviewed to summarize the challenge in the field. Why the proposed method can help to solve the challenge should be explained.

Response 1: In the abstract, previous works on the topic of image authentication have been added to summarize the challenge in the field. The added and modified parts are marked in red font. The experimental results verify the effectiveness of the method.

 Point 2: More previous works on image authentication should be presented in the introduction part, and should be analyzed and compared in the experimental section.

Response 2: Some previous works on image authentication have been presented in the introduction part. In the experimental results part, two comparative line diagrams (FIG. 16 and FIG. 17) are added which better verify the robustness of the method in this paper. The added and modified parts are marked in red font.

Point 3: Equations should appear as part of a sentence and should be punctuated accordingly. However, most of the equations presented in this manuscript are not treated as part of a sentence. The unprofessional mathematical writing significantly degrades the quality of this manuscript and leads to an unpleasant reading experience.

Response 3: In order to improve readability, some mathematical equations have been revised. All the equations presented in this manuscript are treated as part of a sentence. The added and modified parts are marked in red font.

 Point 4: The format of the reference is not consistent.

Response 4: All references have been revised according to the requirements of unified format.

Point 5: My major concern is the novelty of this paper. It appears to me the contributions in this paper are existing well-established methods exploited in authentication. The novelty of the methodology is limited. The authors should justify the originality of the proposed method.

Response 5: By highlighting the visual quality of medical images and using the energy entropy of the time-frequency data as the feature of authentication information.The proposed authentication method combines the advantages of multi-resolution analysis of wavelet packet decomposition and the stability of local energy entropy. No noise is added in the authentication process and the integrity goal of the medical information authentication process is achieved.

 Point 6: The experimental results section should be well presented to show how the parameters affect the performance of the proposed method.

Response 6: In order to better describe the comparison results two comparison graphs are given in Fig. 16 and Fig. 17.

Thanks again for your hard work.

Reviewer 3 Report

The authors proposed a medical image authentication method based on the wavelet packet and energy entropy, and showed by experiments that the proposed method is more robust than the previous methods under multiple types of attacks. But some critical details were missing.

(1) The literature review is weak. The authors compared the proposed method with previous methods in the experiment. However, there was no sufficient discussion on those methods in the introduction. Furthermore, the discussion of similar methods was also missing. Therefore, the innovation and contribution of this paper cannot be justified.

(2) The subsection of the basic theory of wavelet packet transform is not clear. Please first clearly define notations and then use them. The connections between (3) and (4-5) were missing.

(3) Give formal algorithms with pseudo-codes to describe the wavelet packet decomposition of images, ownership construction phase, and ownership verification phase.

(4) In the correlation test between different features, how did you measure the correlation? Please specify it.

(5) The authors showed that the proposed method is more robust in most cases but did not give even intuitive explanations on the result. There are also some cases where the previous methods are better. Please try to give explanations as well.

(6) The writing has to be improved. There are some sentences are too long and each consists of actually two or three sentences without even a comma.

Author Response

Point 1: The literature review is weak. The authors compared the proposed method with previous methods in the experiment. However, there was no sufficient discussion on those methods in the introduction. Furthermore, the discussion of similar methods was also missing. Therefore, the innovation and contribution of this paper cannot be justified.

 Response 1: Some previous works on image authentication have been presented in the introduction part. The added and modified parts are marked in red font.

 Point 2: The subsection of the basic theory of wavelet packet transform is not clear. Please first clearly define notations and then use them. The connections between (3) and (4-5) were missing.

Response 2: The subsection of the basic theory of wavelet packet transform has been rewritten. The added and modified parts are marked in red font.

Point 3: Give formal algorithms with pseudo-codes to describe the wavelet packet decomposition of images, ownership construction phase, and ownership verification phase.

Response 3: The wavelet packet decomposition of images is described as the pseudo-codes. The added and modified parts are marked in red font.

 Point 4: In the correlation test between different features, how did you measure the correlation? Please specify it.

Response 4: The correlation test between different features is evaluated by the normalized similarity value (NC) . The NC value is between 0 and 1. The larger the NC value the higher the similarity between them. The added and modified parts are marked in red font.

Point 5: The authors showed that the proposed method is more robust in most cases but did not give even intuitive explanations on the result. There are also some cases where the previous methods are better. Please try to give explanations as well.

Response 5: In the experimental results part, two comparative line diagrams (FIG. 16 and FIG. 17) are added which better give intuitive explanations on the result. The added and modified parts are marked in red font.

 Point 6: The writing has to be improved. There are some sentences are too long and each consists of actually two or three sentences without even a comma.

Response 6: The writing has been improved and some long sentences have been revised. The added and modified parts are marked in red font.

Thanks again for your hard work.

Round 2

Reviewer 1 Report

I am sorry that my remarks to Version 1 of the paper were not clear enough. Two main problems remain unsolved; both require rejecting the paper as it is.

1) What is exactly the problem the paper tries to solve? In particular, what does "authentication" in the paper title mean?

The problem the paper is about should be clearly stated not later than in the first paragraph of the Introduction. So what is it?  Is it simply connecting the medical image to a person (digital signature)? Making sure that the image is not fabricated (authenticity)? Or is it ensuring that the image during transmission is not manipulated (integrity)? These tasks and their combinations have industry standard cryptographic solutions. Reading further the paper, it says something about manipulating the image. Why is that a concern? Changing a single pixel should invalidate the image -- those cryptography standards take care of this case. So what is the problem for which the above mentioned crypto methods do not work? The Introduction does not give any clue whatsoever what the rest of the paper tries to solve. Why extracting robust information from the image does solve any of the problems listed in the Introduction?

2) Section 2 (titled "Basic theory") tries to give a brief introduction to what wavelet decomposition is. It is messy, apart from repeating some formulas from a technical manual that makes no sense. The wavelet is a hierarchical self-similar decomposition. This section should be an understandable summary for those who are not experts in this particular image manipulating method.

Using wavelet decomposition requires defining the initial wavelets. Without explicitly defining them, there is no way to repeat the reported computation thus making the paper ineligible for any scientific evaluation. The usage of the "entropy" at a certain point is totally arbitrary. Determining a cut-off threshold can be done in other ways; even in the literature there are five other entropy-like amounts discussed for the same purpose. There should be a lengthy discussion of that particular choice.

In summary, this reviewer cannot consider the paper as formally acceptable for publication.

Reviewer 2 Report

I am satisfied with this revision.

Reviewer 3 Report

Compared with the previous version, the presentation of this manuscript has been improved. But the literature review is still weak. In the experiment, many previous algorithms have been compared with the proposed method. But only references [3], [8], [9] are briefly discussed in the introduction. In the revised manuscript, Figures 16, 17 are added to compare the robustness. But the authors only present the results without intuitively explaining why the proposed method is better in most experiments and worse in some cases. There are some typos, e.g., in Line 11, Line 65.